# Airborne LiDAR Point Cloud Classification Using Ensemble Learning for DEM Generation

**DOI:** 10.3390/s24216858

**Published:** 2024-10-25

**Authors:** Ting-Shu Ciou, Chao-Hung Lin, Chi-Kuei Wang

**Affiliations:** Department of Geomatics, National Cheng Kung University, Tainan 70101, Taiwan; p66101079@gs.ncku.edu.tw (T.-S.C.);

**Keywords:** point cloud segmentation, deep learning, DEM generation

## Abstract

Airborne laser scanning (ALS) point clouds have emerged as a predominant data source for the generation of digital elevation models (DEM) in recent years. Traditionally, the generation of DEM using ALS point clouds involves the steps of point cloud classification or ground point filtering to extract ground points and labor-intensive post-processing to correct the misclassified ground points. The current deep learning techniques leverage the ability of geometric recognition for ground point classification. However, the deep learning classifiers are generally trained using 3D point clouds with simple geometric terrains, which decrease the performance of model inferencing. In this study, a point-based deep learning model with boosting ensemble learning and a set of geometric features as the model inputs is proposed. With the ensemble learning strategy, this study integrates specialized ground point classifiers designed for different terrains to boost classification robustness and accuracy. In experiments, ALS point clouds containing various terrains were used to evaluate the feasibility of the proposed method. The results demonstrated that the proposed method can improve the point cloud classification and the quality of generated DEMs. The classification accuracy and F1 score are improved from 80.9% to 92.2%, and 82.2% to 94.2%, respectively, by using the proposed methods. In addition, the DEM generation error, in terms of mean squared error (RMSE), is reduced from 0.318–1.362 m to 0.273–1.032 m by using the proposed ensemble learning.

## 1. Introduction

Digital elevation models (DEM) provide fundamental geographic information for a variety of applications, including national infrastructure planning, land management, and satellite image orthorectification. Airborne LiDAR point cloud data represent one of the major data sources for high-resolution DEM generation. To semi-automatically, or even fully automatically generate DEM, supervised classifiers are typically adopted to extract ground points from a point cloud [1,2], followed by manual editing as a post-processing step to fix the misclassified points. While the recent learning-based studies on 3D point cloud recognition and classification have excellent performance, the methods were evaluated using well-processed LiDAR point clouds containing topographies with simple geometries. The airborne LiDAR point cloud classification for DEM generation is still a challenge for terrains with complicated geometrics.

Early research on DEM generation focused on the extraction of ground points using machine learning techniques [3,4,5]. To filter or classify ground points, these studies typically emphasized the application of handcrafted features, such as eigenvalue-based features derived from point clouds, point distribution, and point density. These features were then utilized in classical machine learning algorithms, including support vector machines [6], random forests, conditional random fields [7], and the cloth simulation filter (CSF) [8,9]. For example, Roberts et al. [10] compared several ground point classifications and revealed the strength of each method. These compared methods aimed to enhance the accuracy and efficiency of ground point extraction for DEM generation by removing the non-ground points.

In recent years, due to the rapid advancement of big data analysis and GPU computing, deep learning techniques have had a significant impact on classification and object recognition, especially for airborne LiDAR point clouds [11,12]. A point cloud classification addresses on classifying each point in a dataset, assigning them to the category of ground point or non-ground point. With the development of deep learning, learning-based ground point classifications have demonstrated superiority over traditional algorithms [13,14,15,16]. For example, the study [17] concluded that PointCNN is better than CSF and Progressive Morphological Filter (PMF) because the traditional algorithms are limited by the high vegetation density of the data, while the deep learning classification is adaptable to complex terrains.

In the realm of point cloud classification using deep learning, point-based and projection-based methods are the major methods to extract features from point clouds and classify them as ground and non-ground points. Projection-based methods compute point elevations, point intensities, and other geometric attributes from 3D point clouds and orthogonally project this information onto an xy-plane to generate feature images. The feature images serve as inputs of a deep learning classification model. The 2D classification results are subsequently back-projected to the original 3D point cloud [11,18,19]. These methods relied on simple projections, that is, transforming 3D point clouds into 2D images. This projection allows the utilization of 2D convolutional neural networks (CNNs), leveraging their success in image classification tasks. However, single-view projections suffer the problem of point cloud geometry degradation because of the 2D projection. To alleviate this problem, sophisticated projection techniques were developed such as multi-view projection. Multi-view projection generates 2D projections or profiles from point cloud data at multiple angles [20,21,22,23]. The classification results obtained using 2D CNNs are then back-projected onto the original point cloud. This advancement marked a significant improvement over the single-view projection techniques.

Point-based methods directly operate on raw point clouds, preserving the inherent geometric and spatial relationships among points [24]. This approach addresses the major difficulty in point cloud classification, that is, the unordered nature of point clouds [25,26]. To tackle this challenge, two main model categories have emerged: employing multilayer perceptrons (MLP) with max pooling and utilizing graph-based structures for feature extraction. PointNet [25] introduced an MLP-max-pooling architecture, which processes each point independently through shared MLPs followed by a global pooling operation to maintain permutation invariance. Despite its innovation, PointNet did not fully consider local features in classification. To address this problem, several researchers have made improvements based on the PointNet framework. Yousefhussien et al. [27] enhanced the PointNet architecture by increasing the dimensionality of feature spaces, which enables an end-to-end classification. Additionally, Li et al. [28] applied a shared MLP structure for feature extraction, specifically focusing on the elevation geometry features of airborne LiDAR point clouds. Furthermore, PointNet++ [29] was developed with a hierarchical design that extracts local structures through nested regions, thereby enhancing its ability to capture fine-grained geometric details in point cloud classification. Concurrently, graph-based methods convert unordered point clouds into graph structures to better model local neighborhood relationships [30,31,32]. Among these, the dynamic graph convolutional neural network (DGCNN) demonstrated superior performance by dynamically updating graph connections to capture intricate local features [33,34].

While the point-based methods excel in retaining spatial information and avoiding the preprocessing step of projection or voxelization, the methods generally suffer high computational costs from the large amount of points. Additionally, the design of effective network architectures for point clouds demands specialized knowledge, presenting a steeper learning curve compared to projection-based methods. Nonetheless, the ability to directly leverage raw 3D data makes point-based methods particularly appealing for applications that require the recognition of object geometrics. Note that many related studies trained and evaluated models using benchmark datasets, such as the ISPRS 3D Semantic Labeling Benchmark and Actueel Hoogtebestand Nederland (AHN), which exhibit stable point cloud quality and relatively simple terrain characteristics. However, the airborne LiDAR point clouds generally exhibit diverse and complex terrain features, including significant elevation variations, and mixed residential and mountain areas. The diverse and intricate characteristics inherent in such LiDAR data will decrease the robustness and classification accuracy of classifiers proposed in related works. Therefore, this study aims to develop point cloud classification models for ground point extraction and DEM generation using deep learning. Instead of presenting a new deep learning model, a model training scheme with geometric features for LiDAR point cloud classification is proposed. First, several specialized models are trained with a set of geometry features, including local point coordinates, local roughness, normalized z and local elevation difference, for specific terrain characteristics. The ensemble learning strategy [35,36] is then applied to combine the specialized point cloud classification model for the separation of ground points from point clouds with complicated topographies.

## 2. Methodology

To achieve the generation of DEM from ALS point cloud, a robust point cloud classifier capable of distinguishing between ground and non-ground points is indispensable. This study aims to develop a deep learning model that classifies point clouds and extracts ground points from ALS data, even for data containing complicated topographies. In this section, the proposed method is introduced, in which the network structure, input features, ensemble learning, and the dataset used for model training and validation are provided. 

### 2.1. Point Cloud Classification Model

Wang et al. [30] introduced a deep learning model called DGCNN, which leverages a graph-based network to extract geometric features from point clouds. The DGCNN encompasses both point cloud classification and segmentation. Only the network of the segmentation model is depicted in Figure 1 since this study addresses segmentation and classification. The input point cloud is organized using a graph-based structure that incorporates edge convolution. Unlike the models based on multilayer perceptrons (MLPs), edge convolution emphasizes geometric relationships among points, which enables the DGCNN to extract local geometric features from point clouds. The input to the model is a set of 3D points of the size  n, and each point contains *m* features. The n×m input tensor undergoes a spatial transformation layer, converting the point cloud into a feature space. The operation of edge convolution is illustrated in Figure 2. In the figure, $\left\{x_{j_{1}},\cdots ,x_{j_{5}}\right\}$ represents the neighboring points of the center point $x_{i}$. $\left\{e_{{ij}_{1}},\cdots ,e_{{ij}_{5}}\right\}$ denotes the edges between $x_{i}$ and $\left\{x_{j_{1}},\cdots ,x_{j_{5}}\right\}$. In edge convolution, a fully connected layer computes the edge features between a central point and its neighboring points. In DGCNN, edge feature aggregation combines the features of all edges connected to a central vertex. For each central point, the new feature is derived by aggregating the edge features from its *k* neighbors. Repeated edge convolution layers that transform the input information to a feature space are used in the neural network design to extract high-level local geometric features from the point cloud. These local features are then aggregated into a global feature, and the output score is computed based on both local and global features.

### 2.2. Model Input Features

In point cloud segmentation/classification models, the input typically consists of point coordinates and intensities extracted from ALS data [13,37]. However, the LiDAR data may be acquired using different LiDAR sensors on different acquisition dates, causing inconsistent intensities of 3D points, as shown in Figure 3. These variations could introduce noise, potentially disrupting the classifier’s accuracy in categorizing the data. Therefore, the point intensity is inappropriate to be regarded as the model input feature. To train a classifier with satisfying performance, alternative data sources or geometric features must be utilized in addition to the point coordinates. In this study, the features of local point coordinate, local roughness, and local elevation difference are introduced and utilized as the model input features. These features are described below.

*Local point coordinate*. Input data normalization is a fundamental step widely utilized in supervised learning, in which the input data are rescaled to a specific range for the mitigation of data noise during training. However, the normalization of point coordinates will remove the scale factor in data, which may reduce the model recognition ability for ground objects with different scales. Utilizing the original coordinates of point clouds as the model inputs can avoid the problem. However, the point coordinates are represented by a geographic coordinate system such as the World Geodetic System 1984 (WGS 84). Using the world coordinate with a large value range to represent 3D points may make the model training difficult to converge. To address this issue, a local point coordinate system is adopted. Specifically, the origin of the local coordinate system is defined as the means of the *x* and *y* coordinates, and the *z* is shifted by the minimum of the *z* coordinate. The local coordinate ($x_{i}^{l}$,$y_{i}^{l}$,$z_{i}^{l}$) of the point i is defined as
(1)$$\left\{\begin{array}{l} x_{i}^{l}=x_{i}-\bar{X}\;\\ y_{i}^{l}=y_{i}-\bar{Y}\\ z_{i}^{l}=z_{i}-\min\left(Z\right) \end{array}\right.,i=1\cdots n,$$
where $\bar{X}$ and $\bar{Y}$ represent the averages of x and y coordinates, min⁡(Z) represents the minimum of points’ z coordinates, and n denotes the number of points. The local coordinates of points ($x^{l},y^{l},z^{l}$) are one of the model inputs. 

*Local roughness*. Several studies demonstrated that the utilization of elevation information in point clouds can benefit the extraction of ground points [13,14]. This is particularly relevant in mountain areas where the ground surface is partially obstructed by the tree canopy, making the identification of ground points solely based on point local coordinates a challenging task. Tin houses with flat rooftops, surrounded by trees and rugged terrain, further complicate the classifier’s performance, often leading to the misclassification of tin house rooftops as ground points. Thus, developing a feature that effectively represents these subtle elevation features is essential for the model training by quantifying and measuring roughness, which represents the distance between each point and the best-fitting plane of its nearest neighbors. The optimal fitting plane is defined as:(2)$$P^{*}\left(p_{i}\right)=arg\;\underset{a,b,c,d}{\min}\sum_{p_{j}:\left(x_{j},y_{j},z_{j}\right)\in N\left(p_{i}\right)}{\left(z_{i}-P\left(x_{j},y_{j},z_{i}\right)\right)}^{2},$$
where $P\left(x_{j},y_{j},z_{i}\right)=ax_{j}+by_{j}+cz_{i}+d$ and $P^{*}(p_{i})$ denotes the optimal fitting plane of the point $p_{i}:(x_{i},y_{i},z_{i})$. $P^{*}(p_{i})$ is calculated by using the neighboring points $N(p_{i})$. With the optimal fitting plane, the roughness is formulated as
(3)$$r(p_{i})=Dist\left(p_{i},P^{*}(x_{i},y_{i},z_{i})\right),$$
where $r(p_{i})$ denotes the roughness of the point $p_{i}$, which is derived from the distance between the point and the fitting plane $P^{*}$. In this study, a point neighborhood radius of 10 m, fully or partially covering the ground objects, was utilized in the implementation. This parameter is tunable, and its setting depends on the sizes of geometric objects. During model training, a point cloud is partitioned into small point sets because of the limited GPU memories. The local roughness for each point is calculated in the original point cloud rather than the partitioned point set by taking the boundary effect of the partitioned point set into account. 

*Normalized z and local elevation difference*. In the ensemble strategy, which will be described in Section 2.3, the geometric differences between urban and mountainous terrains must be reflected in the selected input features, as these terrains are characterized by variations in elevation. Due to the importance of elevation data, the z-coordinate is initially chosen as the primary feature. However, unlike training a ground point classifier, elevation information for individual points is less critical for topographic classifier training. Instead, the global elevation trends within the point clouds are crucial for accurate topography identification. Consequently, normalization is applied to the z coordinates, thereby defining the feature of normalized z as:(4)$${nz}_{i}=\frac{z_{i}-\bar{Z}}{\sigma (Z)},i=1\cdots n$$
where $z_{i}$ represents the elevation value of the point *i*; $\bar{Z}$ and σ(Z) denote the average and standard deviation of the elevations of points, respectively. 

Elevation information is crucial for topographic classification, though it does not necessitate the detailed point cloud features required for ground point classification, such as the relationships between individual points and the overall shape and structure of the point cloud. Instead, topographic classification emphasizes elevation characteristics. Urban areas generally exhibit flat terrain, whereas forest areas have more varied and undulating landscapes with dispersed elevation distributions. Consequently, the differences between individual points and the average elevation of neighboring points become key features for distinguishing between urban and forest topographies. Consequently, a feature named local elevation difference (denoted as $\;{∆z}_{i}$) is used. In this study, the local elevation difference is computed as the difference between the elevation of individual points and the average elevation of their neighboring points, denoted as:(5)$${∆z}_{i}={\bar{Z}}_{r}-z_{i},\;i=1\cdots n,$$
where ${\bar{Z}}_{r}$ stands for the average elevation value of all points within a radius of r meters. To preserve the inherent features within the subsets while capturing significant traits within a search area equivalent to an average building’s size, the value of r is set as 10 m in this research.

### 2.3. Ensemble Learning

Ensemble learning is a powerful machine learning technique that boosts classification performance by using multiple models. The fundamental concept behind ensemble learning is constructing a set of base models that jointly generate inferencing results for a problem. These base models can be trained using different methodologies, architectures, and training datasets. The primary ensemble learning approaches include bagging, boosting, and stacking [36]. The core principle of boosting [35] involves combining several relatively weak base models to improve inference results, achieved by iteratively adjusting the weights of each base model to enhance overall accuracy [38]. With the concept of boosting strategy, this study employs ensemble learning to integrate two specialized ground point classifiers designed for different terrains, capitalizing on the feature extraction capabilities of DGCNN. This method aims to enhance the accuracy of ground point classification, delivering high-precision results in point cloud data with complicated terrain. Topographical characteristics in most landscapes are varied. Urban regions mainly consist of flat landscapes and artificial buildings, while mountain regions predominantly exhibit rough terrain. The urban and mountain areas display different topographical characteristics in terms of relative elevations and the aspect of the surface. To ease the difficulty of LiDAR point cloud classification, the classifiers and topographies are separated into two types, that is, urban and mountain, under the scheme of boosting ensemble learning. Consequently, the classifiers, named $M_{urban}$ and $M_{mountain}$, can address urban and mountain point clouds, respectively, and effectively learn the ground point features during model training. However, when a point set contains both urban and mountain topographies, the challenge of utilizing a single classifier for ground point extraction arises. Therefore, a topography classifier is necessary to recognize the local topography and integrate the classification results of two specialized classifiers. A DGCNN-based topography classifier $M_{t}$ is trained for the semantic segmentation of urban and mountain regions from a point cloud. The topography classifier $M_{t}$ is used to provide the urban and mountain probabilities for the models $M_{urban}$ and $M_{mountain}$ under the ensemble learning scheme. The model $M_{t}$ takes local point cloud coordinates, local elevation differences, and normalized elevation as the input features, denoted as $f_{t}=(x^{l},y^{l},z^{l},nz,∆z)$. The model outputs the probabilities of each point belonging to urban topography, denoted as $W_{u}$. The probabilities of each point belonging to mountain topography is obtained by $W_{m}=1-W_{u}$. These two probabilities serve as weights for aggregating the outputs of the models $M_{urban}$ and $M_{mountain}$.

In data referencing, the local point coordinate and local roughness are taken as the input features, denoted as f = $\left(x^{l},y^{l},z^{l},r\right)$ for the models $M_{urban}$ and $M_{mountain}$. The classifiers $M_{urban}$ and $M_{mountain}$ predict the probability of each point belonging to ground category, denoted as $P_{u}^{gd}$ and $P_{m}^{gd}$, respectively. And, the probabilities of non-ground for the classifiers $M_{urban}$ and $M_{mountain}$ can be obtained by $P_{u}^{nongd}=(1-P_{u}^{gd})$ and $P_{m}^{nongd}=(1-P_{m}^{gd})$, respectively. A weighted technique is used in this work to determine the categorization probabilities for each location as either ground or non-ground. The calculation is shown in Equation (6):(6)$$\left\{\begin{array}{cc} P^{gd}= & P_{u}^{gd}\times W_{u}+P_{m}^{gd}\times W_{m}\\ P^{nongd}= & (1-P_{u}^{gd})\times W_{u}+(1-P_{m}^{gd})\times W_{m} \end{array}\right.,$$
where $P^{gd}$ and $P^{nongd}$ represent the predicted probability values for ground and non-ground point categories, respectively. After determining the probability values for each point’s classification as either ground or non-ground, each point is then allocated to the category associated with the higher probability value. The workflow of ground point determination by using ensemble learning technique is demonstrated in Figure 4, and the model with ensemble learning is denoted as $M_{ensem}$.

### 2.4. Training Dataset and Data Preprocessing

The point clouds mainly used in this study originates from a Taiwan LiDAR surveying project conducted between 2016 and 2020. Point cloud selection is based on penetration, calculated in 10 × 10 m units. The selected data must ensure that over 70% of the cells exhibit a penetration rate exceeding 50%, with each point cloud containing around 9 million points. The locations of selected point clouds are shown in Figure 5. The labels of the point clouds are labeled manually. The training dataset comprises 3D point coordinates, point intensities, and labeled classes. To enhance training stability, the point clouds in the dataset are categorized into urban, mountain, and mixed-topography subsets, depending on the underlying topographical characteristics. The urban subset comprises point clouds associated with flat terrains and artificial structures such as buildings, bridges, and highways. In contrast, the mountain subset encompasses terrains with significant elevation variations, including mountainous and hilly regions. Additionally, the mixed-topography subset constructed from the point cloud contains both urban and forest terrain features. During the training of the urban and mountain classifiers $M_{urban}$ and $M_{mountain}$, the corresponding training subsets are utilized. In the training of the topography classifier $M_{t}$, both the urban and mountain subsets are employed. The labels ‘urban’ or ‘mountain’ are assigned to points based on their topographical characteristics.

The input tensor size of the DGCNN is fixed, which requires preprocessing to partition the point cloud into several subsets of the input size *n* for model training and inferencing. An irregular partition is performed with a dynamic window, and then a sampling process is performed for each partitioned set to meet the requirement of the fixed input tensor and to preserve point geometric features. The partition and sampling procedures have been tailored to meet the requirements of model training and inferencing, respectively. For the model training, the primary objective is to facilitate the comprehensive learning of geometric features. The procedures of point cloud partition and sampling are designed to maximize the description of the geometric features of partitioned sets and increase the diversities of the training set. To achieve these goals, the location of a partition window is randomly selected in the xy-plane, and the size of the partition window is dynamically adjusted based on the number of points contained in the window. On the other hand, the window size will be gradually enlarged until the number of the contained points is larger than *n*. In point sampling, the farthest point sampling (FPS) [39] which can efficiently retain point geometric features is adopted. Each input point cloud is preprocessed using the mentioned partition and sampling processes and result in 1000 separated point subsets for model training. For model inferencing, the major objective is applying the trained model to inference all the points in a point cloud. With the objective, the point cloud is segmented into several disjoint subsets using a xy-grid-based partition. The size of the grid is determined by using the average size of the partition windows during training. Note that this partition cannot ensure a fixed number of points in each subset. As a result, the sampling process evolves dynamically in response to the amount of points in each subset. When a subset has an inadequate number of points, a random up-sampling step is adopted to augment the number of points until it matches the required quantity, that is, *n*. Conversely, if a subset contains an excessive number of points, the FPS is utilized to reduce the density of the point subset. This approach ensures that the classification can be performed on each point while the point density of the subsets can be managed with the consideration of well model performance.

To bolster the credibility of the proposed method, the open-access AHN (Actueel Hoogtebestand Nederland) dataset was also utilized [40]. This high-resolution LiDAR-based dataset for the Netherlands features point clouds with an average density of 10 points per square meter and a vertical accuracy between 0.15 and 0.2 m. Each point is labeled into one of five classes: ground, water, bridge, building, and vegetation. For the purposes of this study, all non-ground labels were merged into a single “non-ground” class. Additionally, three point clouds with terrain characteristics closely matching the experimental requirements were selected from the AHN dataset for testing. 

## 3. Experimental Results and Discussion

This section presents the experimental results and the improvements achieved by the proposed ensemble learning. The proposed method involves the classification of point clouds and the determination of ground points by combining the outputs of the models $M_{urban}$ and $M_{mountain}$ with the weights from the topography classifier $M_{t}$. Each classifier is trained by using its specialized dataset. The evaluation experiments for the proposed ensemble learning method were conducted on an NVIDIA GeForce RTX 3090 GPU, resulting in an average execution time of approximately 260.36 s and a memory usage of around 4333.42 MB. This computational cost is relatively manageable compared to the time and effort typically required for traditional DEM generation processes, making it a more practical and efficient alternative. The classification results are analyzed in 3D-point and DEM domains, which are described in Section 3.1 and Section 3.2, respectively. The evaluations are conducted using the root mean square error (RMSE) between the generated and the reference DEM.

### 3.1. Evaluation in 3D Point Domain

The proposed ensemble learning method, denoted as $M_{ensem}$, contains two classifiers, $M_{urban}$ and $M_{mountain},$ for ground point extraction and a topography classifier $M_{t}$ for topography type determination. The validation experiments were conducted to compare the models $M_{urban}$, $M_{mountain}$, and $M_{ensem}$ by using the validation dataset. The confusion matrices shown in Table 1 are used to present the performance of the classifiers on the validation dataset. To validate the models $M_{mountain}$ and $M_{urban}$, the classification results underscore the strengths of these specialized models. The model $M_{urban}$ demonstrates high accuracy on urban datasets, particularly in predicting non-ground points derived from man-made structures, as shown in Figure 6. The confusion matrix indicates that the model $M_{urban}$ achieves approximately 2% higher accuracy on urban datasets compared to its performance on mountain datasets. Conversely, the model $M_{mountain}$ shows superior accuracy on mountain datasets, excelling in identifying ground points with significant elevation disparities within the point cloud, as shown in Figure 7. 

However, the classification accuracies decrease when applying the models $M_{urban}\;$ and $M_{mountain}$ to the mixed dataset. To evaluate the effectiveness of the proposed ensemble learning method, scenes containing multiple terrains were tested by using the models $M_{urban}$, $M_{mountain}$ and $M_{ensem}$. The prediction results are shown in Figure 8. The first three scenes contain multiple terrain types, including mountain and artificial buildings, and the last scene is the suburban type.

The results show that the model $M_{urban}$ has better performance on the man-made structures than the model $M_{mountain}$. Conversely, the model $M_{mountain}$ provides higher precision in the ground point determination in mountain datasets compared to the model $M_{urban}$. In comparison, the proposed model $M_{ensem}$ accurately predicts both ground and non-ground points from point clouds with either mountain or urban features. The confusion matrix also reveals that the model $M_{ensem}$ effectively reduces the misclassification of non-ground points, thereby improving their classification accuracy by 2%. This suggests that the model $M_{ensem}$ capitalizes on the strengths of both $M_{urban}$ and $M_{mountain}$, ultimately enhancing the overall accuracy of ground point classification in point cloud data with complicated topography, as shown in Table 2. 

In conclusion, the model $M_{ensem}$ offers substantial benefits over specialized models in the classification of point clouds with varied and intricate topographies. In Table 3, the classification results from DGCNN, PointNet, and our proposed method are compared. In this comparison, our proposed method provides an improvement on the LiDAR point cloud classification compared with DGCNN and PointNet. The design of the graph network in DGCNN improves the overall performance of classification on the Taiwan dataset compared to the PointNet dataset, and the proposed boosting ensemble scheme further improves the classification accuracy. Additionally, the comparison of the AHN dataset suggests that the proposed method can be broadly applied to other datasets, demonstrating similar classification capabilities that could help optimize the DEM generation processes (Figure 9). This comparison further improves the proposed approach, which not only improves classification accuracy through the use of a graph structure network but also leverages the strengths of specialized classifiers tailored to different topographic features. The integration of these advancements within the boosting ensemble learning strategy significantly enhances overall performance in diverse terrain classifications.

### 3.2. Evaluation of DEM Domain

The DEM accuracy in the evaluation of point cloud classification can be discussed from two perspectives. First, not all points containing ground point features will be assigned the ‘ground point’ label. The ultimate purpose of assigning the ground point label is to generate a DEM through interpolation. Consequently, if a point is identified as representing the ground, other points with similar features are not labeled as ground points and are instead labeled as non-ground. While this process ensures precise ground point selection, it also means that some non-ground points, even those with ground-like features, may be labeled as non-ground. Such labeling strategy can lead to inaccuracies in the accuracy index, affecting the classifier’s performance assessment. Figure 10 illustrates a situation where the classifier classifies more ground points than the labeled data, yet the overall shape of the DEM remains relatively unchanged. Secondly, given that the main objective of point cloud classification is to accelerate the post-processing stage of DEM development, the accuracy in 3D point domain only reflects the number of correctly predicted points in the classification results. However, this does not ensure the overall structural classification accuracy of the objects. In other words, classification accuracy cannot verify whether the classifier can entirely predict an entire man-made structure as non-ground points. Therefore, it is essential to take the accuracy of the DEM produced based on the classification results into account as well. 

To assess the accuracy of the DEM, this study examined the elevation discrepancies between the DEM generated from three distinct point cloud classification methods and the ground truth DEM. The evaluation of three ground point classifiers is performed on point cloud data from the mixed dataset that was not included in the training process. The terrain feature and DEM are demonstrated in the ortho-image column of Figure 11. The statistical results are presented in Table 4, and a visual comparison of elevation and height differences between the DEM generated by the ground point prediction and the ground truth DEM is depicted in Figure 11**.** The disparity between the ground truth DEM and the prediction DEM is depicted through a color-coded visualization. Green signifies a difference within the range of ±0.2 m, red represents a difference exceeding 1 m, and blue corresponds to a difference less than −1 m. Additionally, root mean square error (RMSE) is applied to demonstrate the overall accuracy of DEM. 

Comparing the DEM difference, the error pattern on specialized classifiers exhibited by the difference between the generated DEM and the reference DEM is consistent with the analysis in Section 3.1 (Figure 11). In the differential analysis of the DEM, the DEM obtained from ground points collected by the urban classifier has a sparse error distribution. The red and blue area scattered throughout the image suggests that despite the urban classifier’s strong capacity to detect non-ground features, the complicated topography challenges the classifier’s work of distinguishing between man-made structures and cruel terrain. On the other hand, the DEM generated from ground points detected by the mountain classifier shows a concentrated inaccuracy in the range of more than 1 m when compared to the reference DEM. This shows that the mountain classifier is good at retrieving ground points from wooded regions. However, the classifier has difficulty coping with man-made structures located in cruel terrain. In the DEM obtained from ground points extracted by an ensemble learning process, the result shows that ensemble learning combines the capabilities of both the mountain and urban classifiers has been established. The red and blue area in visualization difference of DEM become less compare to DEM generated with points that extract by using specialized classifier alone, which implies that the ground point in curve terrain could be captured with the ability of mountain classifier, and the non-ground points belongs to the man-made structure could also be recognized with the ability of urban classifier. In comparison to previous approaches, the ensemble learning strategy decreases the misclassification of ground points and increases the accuracy of the DEM, as indicated by a reduced RMSE value. Furthermore, the proposed made the assumption that pixels in the generated DEM with elevation errors falling between ±0.2 m do not need additional manual editing on point cloud classification. The DEM generated from the classification result of ensemble learning strategy requires the least amount of manual editing, according to the three generated DEM results, and the areas that do require manual editing are concentrated in areas with distinct structures, which makes the editing process relatively time-efficient. Overall, these results highlight the potential benefits and difficulties of ensemble learning for DEM generation and point cloud classification.

## 4. Conclusions

This study developed a deep learning-based method for ground point classification in aerial LiDAR point clouds. Owing to Taiwanese point cloud data’s complicity with topography, the application of ensemble learning, which combines a topography classifier with two specialized ground point classifiers made for urban and mountain datasets, allows for the optimization of each ground point classifier’s unique strengths. Our approach has shown enhanced performance in classifying point cloud data with complicated geometric characteristics when compared to existing methods. Additionally, the outcomes show a reliable categorization ability in Taiwan on various terrains. The prosed ensemble learning strategy is model-independent. In the near future, we plan to apply this learning strategy to advanced deep learning point cloud classification and ground filtering models, such as PointTransformer [41] and PointNetV3 [42].

However, the requirement to resample point cloud data to fit the input layer remains a challenge. While the sampling process is necessary due to the structure of the model, it may introduce subtle alterations to local geometric features, potentially impacting feature extraction accuracy. Since raw point clouds generally have varying point numbers, developing a model capable of directly processing these variations would greatly improve the classifier’s practical applicability. This limitation underscores an important avenue for future research, focusing on improving the model’s adaptability to different point cloud densities and ensuring accurate feature extraction, even in the presence of subtle geometric variations. Despite these challenges, this study highlights the potential of the proposed method to contribute to the development of highly accurate DEM.

## Figures and Tables

**Figure 1 sensors-24-06858-f001:**
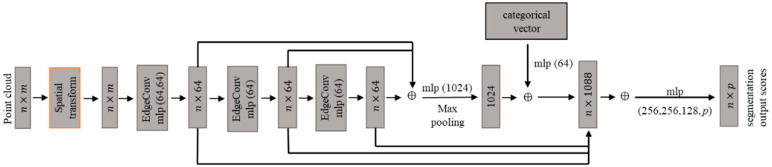
The network structure of the DGCNN segmentation model.

**Figure 2 sensors-24-06858-f002:**
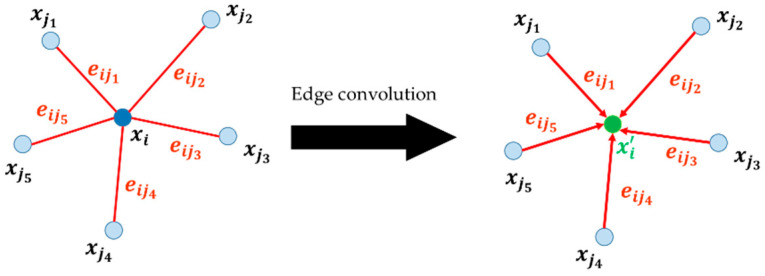
The edge convolution operation.

**Figure 3 sensors-24-06858-f003:**
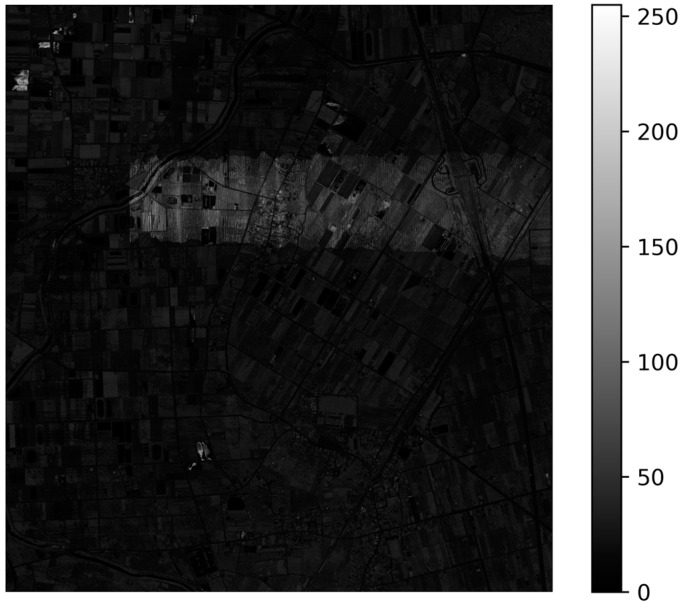
The inconsistent intensity value in point cloud data.

**Figure 4 sensors-24-06858-f004:**
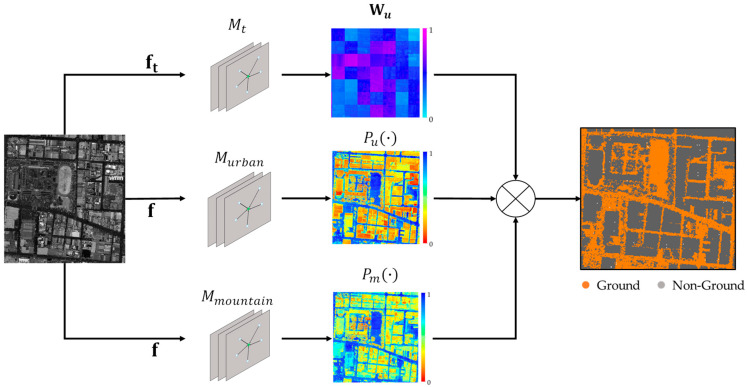
Workflow of ground point determination by using ensemble learning.

**Figure 5 sensors-24-06858-f005:**
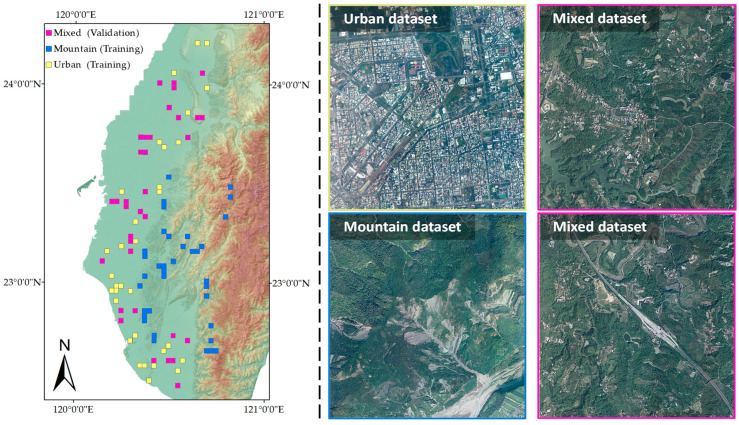
Spatial distribution of training datasets. (**Left**) the locations of mountain, urban, and mixed datasets are marked gray, orange, and pink; (**right**) examples of datasets.

**Figure 6 sensors-24-06858-f006:**
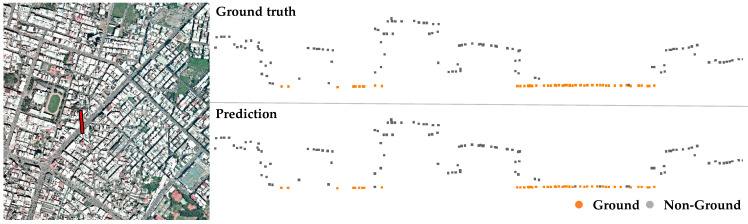
Results of the urban classifier $M_{urban}$ applied to an urban dataset. (**Top**) ground truth; (**bottom**) prediction result. The point cloud profile of the red line in the left subfigure is displayed in the right subfigure, and the ground points are marked in orange.

**Figure 7 sensors-24-06858-f007:**
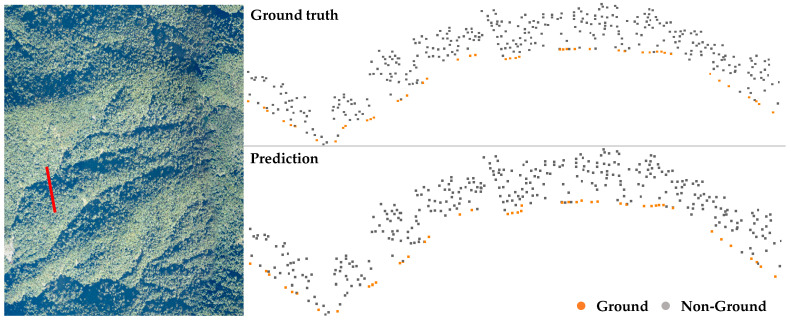
Results of the urban classifier $M_{mountain}$ applied to mountain data. (**Top**) ground truth; (**bottom**) prediction result. The point cloud profile of the red line in the left subfigure is displayed in the right subfigure, and the ground points are marked in orange.

**Figure 8 sensors-24-06858-f008:**
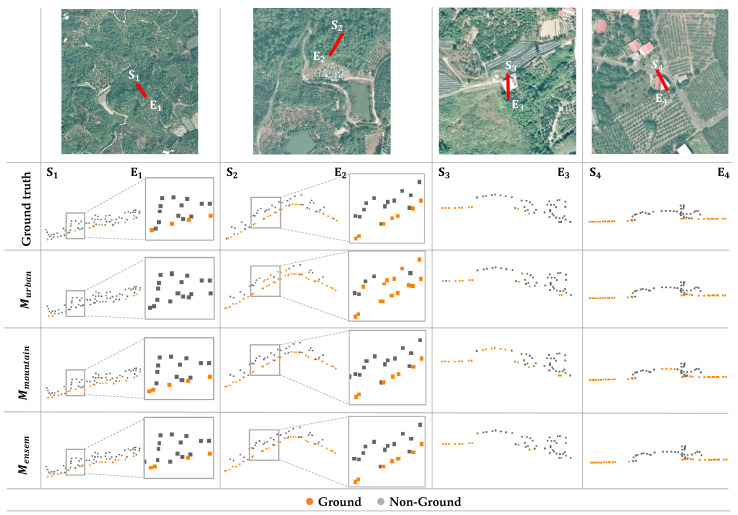
Comparison of prediction results of three ground point extraction processes on the mixed dataset. The $S_{i}$ and $E_{i}$ represent the start and end of the profile. The locations of the profiles are marked with red, and the ground points are marked in orange.

**Figure 9 sensors-24-06858-f009:**
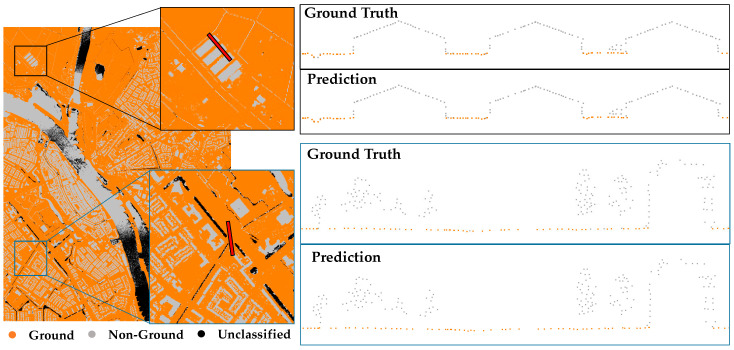
Classification result of AHN dataset using the proposed method. The locations of the profiles are marked with red, and the ground points are marked in orange.

**Figure 10 sensors-24-06858-f010:**
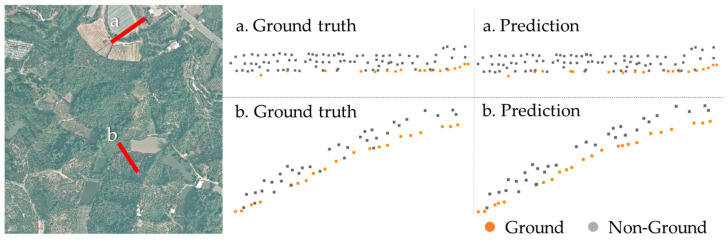
Comparison of ground points in ground truth and prediction. The locations of the profiles are marked with red.

**Figure 11 sensors-24-06858-f011:**
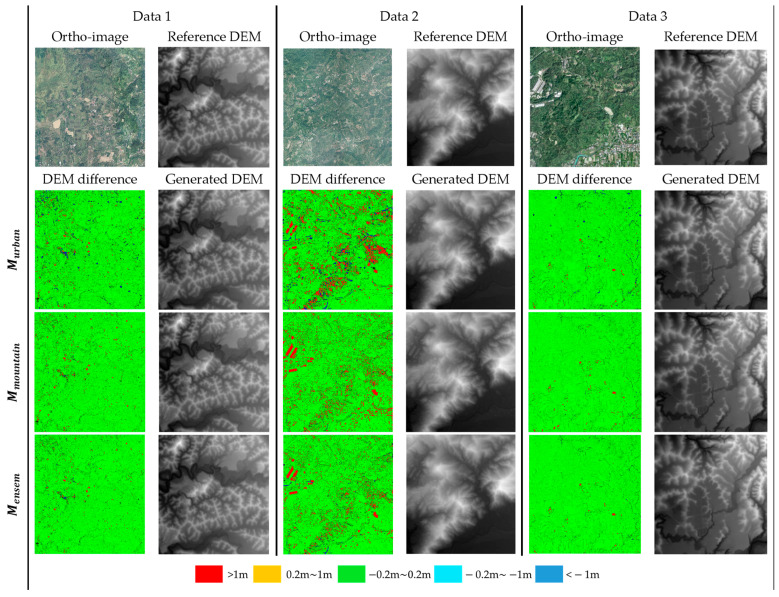
Error maps of the generated DEM.

**Table 1 sensors-24-06858-t001:** Confusion matrix of the point cloud classifiers $M_{urban}$, $M_{mountain}$, and $M_{ensem}$. The urban, mountain, and mixed datasets are tested.

	Ground Truth	$M_{urban}$		$M_{mountain}$		$M_{ensem}$	
Prediction		Ground	Non-Ground	Ground	Non-Ground	Ground	Non-Ground
Urban dataset	Ground	53.0%	7.2%	53.5%	6.8%	54.0%	6.2%
	Non-ground	0.2%	39.5%	0.4%	39.3%	0.2%	39.5%
Mountain dataset	Ground	79.3%	4.9%	80.5%	3.7%	80.6%	3.5%
	Non-ground	4.7%	11.1%	1.6%	14.3%	1.6%	14.2%
Mixed dataset	Ground	59.4%	9.7%	63.5%	5.7%	63.0%	6.2%
	Non-ground	2%	28.8%	1.9%	28.9%	1.7%	29.2%

**Table 2 sensors-24-06858-t002:** Comparison of point cloud classification using the models $M_{urban}$, $M_{mountain}$ and $M_{ensem}$. The measurements accuracy (ACC.), precision (Prec.), recall, and F1 score are used.

Validation Dataset	($\mathrm{Urban}\;\mathrm{Model}\;M_{urban}$)				($\mathrm{Mountain}\;\mathrm{Model}\;M_{mountain}$)				($\mathrm{Ensemble}\;\mathrm{Learning}\;M_{ensem}$)			
	Acc.	Prec.	Recall	F1	Acc.	Prec.	Recall	F1	Acc.	Prec.	Recall	F1
**Urban dataset**	0.926	0.996	0.880	0.935	0.928	0.993	0.887	0.937	**0.936**	0.997	**0.896**	0.944
**Mountain dataset**	0.904	0.944	0.942	0.943	0.948	0.981	0.957	0.969	**0.948**	0.980	**0.958**	0.969
**Mix dataset**	0.883	0.967	0.860	0.910	0.924	0.970	0.918	0.944	**0.922**	0.974	**0.911**	0.942

**Table 3 sensors-24-06858-t003:** Comparison between the proposed model and related models.

Dataset	Model	Accuracy	Precision	Recall	F1 Score
Taiwan LiDAR Dataset	PointNet [22]	0.769	0.704	0.904	0.792
	DGCNN [28]	0.809	0.986	0.704	0.822
	$\mathrm{Ours}\;(M_{ensem}$)	**0.922**	0.974	**0.911**	**0.942**
AHN Dataset	PointNet [22]	0.944	0.950	0.972	0.961
	$\mathrm{Ours}\;(M_{ensem}$)	**0.953**	**0.967**	0.968	**0.968**

**Table 4 sensors-24-06858-t004:** Performance comparison of the models $M_{urban}$, $M_{mountain}$, and $M_{ensem}$, in terms of generated DEM accuracy.

	MModel	Max. (m)	Min. (m)	Difference > 0.2 m (%)	Difference < −0.2 m (%)	RMSE
Data 1	Urban ($M_{urban}$)	9	−11	5.2%	2.2%	0.338
	Mountain ($M_{mountain}$)	10	−11	3.6%	2.1%	0.318
	Ensemble ($M_{ensem}$)	**10**	**−11**	**3.5%**	**2.0%**	**0.273**
Data 2	Urban ($M_{urban}$)	15	−13	6.3%	4.9%	0.480
	Mountain ($M_{mountain}$)	16	−11	5.2%	3.2%	0.393
	Ensemble ($M_{ensem}$)	**10**	**−12**	**4.2%**	**3.3%**	**0.344**
Data 3	Urban ($M_{urban}$)	26	−13	14.4%	7.1%	1.362
	Mountain ($M_{mountain}$)	18	−11	13%	3.7%	1.381
	Ensemble ($M_{ensem}$)	**16**	**−11**	**10.1%**	**4.2%**	**1.032**

## Data Availability

Data are contained within the article.

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
