# Peer review of "Airborne LiDAR Point Cloud Classification Using Ensemble Learning for DEM Generation"

_sensors, 2024, doi:10.3390/s24216858_

Round 1

Reviewer 1 Report (Previous Reviewer 1)

Comments and Suggestions for Authors

Thanks for authors’ detailed revisions and most concerns have been addressed. The revised version adds to the previous manuscript by adding the AHN dataset and comparing it during the testing of the dataset to increase the credibility of the method. In addition, the authors have responded the suggestions.

However, in the part of Figure 8, the regional characteristics of the four scenarios and the reasons for their selection are only stated in the responses and not reflected in the manuscript, which the authors could add to reflect the logical thinking.

You hold that the proposed ensemble learning is point cloud classification and ground filtering models, but in your revised manuscript, ground filtering is still main task, more demonstration or comparison about point cloud classification in your would be advisable.

Author Response

Comment 1. In the part of Figure 8, the regional characteristics of the four scenarios and the reasons for their selection are only stated in the responses and not reflected in the manuscript, which the authors could add to reflect the logical thinking.

Response 1: 

Thanks for the suggestions regarding the description and explanation of Figure 8. In the revision, the reason of choosing these four scenes is provided. For this comment, our revision is described as follow.

3. Experimental Results and Discussion: Page 10, Line 378-383.

However, the classification accuracies decrease when applying the models  and  to the mixed dataset. To evaluate the effectiveness of the proposed ensemble learning method, scenes containing multiple terrains were tested by using the models ,  and . The prediction results are shown in Figure 8. The first three scenes contain multiple terrain types, including mountain and artificial building, and the last scene is suburban type.

Comment 2: You hold that the proposed ensemble learning is point cloud classification and ground filtering models, but in your revised manuscript, ground filtering is still main task, more demonstration or comparison about point cloud classification in your would be advisable.

Response 2: 

Thanks for the advice for the comparison between point cloud classification and ground point filtering. In this work, we adopt a point cloud classification model to separate ground points from LiDAR point cloud data. In our point of view, this process is more like a binary classification task, that is, classifying “ground point” and “non-ground point” from a point cloud. Though there are several differences in point cloud classification and ground point filtering, we have cited papers and references of the two methods with discussion. For this comment, our revision is described as follow.

1. Introduction: Page 1, Line 44-47.

For example, Roberts et al. [10] compared several ground point classifications and revealed the strength of each method. These compared methods aimed to enhance the accuracy and efficiency of ground point extraction for DEM generation with the strategy of removing the non-ground points.

1. Introduction: Page 2, Line 50-57.

A point cloud classification addresses on classifying each point in a dataset, assigning them to the category of ground point or non-ground point. With the development of deep learning, learning-based ground point classifications have demonstrated superiority over traditional algorithms [13, 14, 15, 16]. For example, the study [17] concluded that PointCNN is better than CSF and Progressive Morphological Filter (PMF) because the traditional algorithms are limited by the high vegetation density of the data, while the deep learning classification is adaptable to complex terrains.

Reference: Page  10, 16, 17

[10] K. C. Roberts, J. B. Lindsay and A. A. Berg, " An Analysis of Ground-Point Classifiers for Terrestrial LiDAR," remote sensing, vol. 16, p. 1915, 2019.

[16] P. Zhao, H. Guan, D. Li, Y. YU, H. Wang, K. Gao, J. M. Junior and J. Li, "Airborne multispectral LiDAR point cloud classification with a feature Reasoning-based graph convolution network," International Journal of Applied Earth Observation and Geoinformation, vol. 105, p. 102634, 2021.

[17]NadeemFareed, J. P. Flores and A. K. Das, "Analysis of UAS-LiDARGroundPointsClassification in Agricultural Fields Using Traditional Algorithms and PointCNN," remote sensing, vol. 15, p. 483, 2022.

Reviewer 2 Report (New Reviewer)

Comments and Suggestions for Authors

This paper proposes a deep learning based method that uses ensemble learning strategies to improve the accuracy of ground point classification, which is beneficial for the generation efficiency of DEM. The experiments use LiDAR datasets from Taiwan and publicly available AHN datasets to test the algorithm's generalization ability. However, I still have the following questions:

1. I do not think the ground/non-ground classification of point cloud is a necessary process for DEM generation, which can only improve the speed and automation of DEM generation, so I think the title needs to be modified and qualified.

2. The digital elevation models (DEMs) appearing in the Abstract are identical to the digital elevation models (DEM) at the beginning of the Introduction.

3. Line 30: To semi-automatically generate DEMs, I think this statement is not correct. At present, DEM production is semi-automatic or even fully automated.

4. Line 102 to line 108, this part should express the contribution of this paper, should indicate what kind of modification and innovation in the algorithm, and what kind of specialized point cloud models need to be detailed. Instead of just using "An ensemble learning strategy [30, 31] is applied to combine the specialized point cloud models for ground point extraction. "This expression is passed over.

5. Please introduce the realated work about the classification of point cloud ground/non ground points.

6. Line 134 to line 135: For "repeated block," only the edge convolution layer is repeated in Figure 1.

7. Line 189 to line 190, "The local roughness for each point is calculated in the original point cloud rather than the partitioned point set by taking the boundary effect of the partitioned point set into account. "Are the other parts calculated in blocks or point by point?

8. What are Murban and Mmountain's models like? Only the network architecture shown in Figure 1 is used, which is specially trained for different Urban and Mountain. Or are different strategies or modules designed on the basis of the network structure? What is the originality of the article? Please add explanations where appropriate.

9. Line 274, "and each point cloud containing around 9 million points," is incorrect.

10. Figure 5 is not displayed in the file I downloaded. Please confirm.

11. Table 2 shows the comparison between the proposed algorithm and PointNet and DGCNN. Please add appropriate descriptions in the text.

12. In Table 3, only the experimental results of PointNet are shown. Why did the DGCNN experiment not be conducted?

13. Line 384, "and having a performance that is more suitable for the DEM generation (Figure 9)". How is this statement proved in Figure 9? Not very clear.

14. Line 404 to line 407, 13. Line 404 to line 407, "Consequently, if a selected point accurately represents the ground,  no further points will receive the ground point label and will automatically be classified as non-ground points. This implies that some non-ground points may actually exhibit ground-like features. "What does that mean?

15. The experiment in Figure 9 shows that there will be large errors in mountain DEM generated by Murban, but it seems that there is not much difference between Mmountain and Mensem in intuitive vision. If there are really large errors in some areas, please use the enlarged graph to represent them. If the error is really not large, please consider the experiment of adding an Urban region data set; from the applicability of the proposed method, it can show better performance in both the Urban region and the Mountain region.

Comments on the Quality of English Language

no

Author Response

We would like to give our sincere thanks to the reviewers’ insightful comments and the suggestions to improve the paper. We appreciate all the remarks regarding the presentation and approach description, and are happy to incorporate them into this revised manuscript. The main revisions or explanations to these comments and suggestions are listed below.

  • More related papers on ground/non-ground point classification are cited with discussion.
  • The contribution of this work is further emphasized in the revised manuscript.
  • The descriptions about the experimental results are improved.

Your kind decision on this paper will be very much appreciated. For the point-by-point response please  see the attachment.

This manuscript is a resubmission of an earlier submission. The following is a list of the peer review reports and author responses from that submission.

Round 1

Reviewer 1 Report

Comments and Suggestions for Authors

This manuscript proposes a point-based deep learning model for processing ALS point clouds with complex topography, by involving ensemble learning and a set of geometric features. It has some novelty from technical view, while more comparison is required. Besides, authors didn’t finish a classification work, but actually a ground filtering work. Detailed comments are as follows.

Lines 96-105: The characterisation of the airborne LiDAR point cloud is described but not linked to the following lines to further explain the author's rationale for using it. Also, the later lines introduce the purpose of the research in this manuscript in a rather rigid manner, and the logic of the lines needs to be improved.

Abstract. More contents about the method should be added. Only on sentence (line 14-15) is about method now.

The graphical language of the results of the classifier processing in Figure 5 is not clear enough, and should be preceded by a description of the processing results in the figure as the area drawn by the red line.

Figure 7, why are these four scenes chosen, what are the characteristics of the selected areas? Some of the orthophotos in the profile sections is generally poor displayed; a brief explanation of the reasons for selecting these regions would be advisable.

Table 3: Is the selection of relevant models a bit old? Besides, the proposed method seems a binary division: ground and non-ground. Thus it is actually a ground filtering method and should be also compared with ground filtering methods.

Section 3.2: The layout of the figures in this section is confusing, Figure 9 and Figure 10 are repeated and Figure 10 is not mentioned in the text.And you should check if the section numbers and the figure numbers of this manuscript are correctly ordered.

Line 414, there is no section 3.1.1 in your manuscript.

Section 4: The conclusion section is abbreviated and only summarises the manuscript, the limitations and future research prospects section needs to be added.

Reviewer 2 Report

Comments and Suggestions for Authors

This paper presents a study to identify ground points in a LiDAR 3D point cloud, and then a DEM is generated. The method is based on the application of the DGCNN deep learning method published in 2019. DGCNN works fine in urban areas ad in mountains separately. Authors aggregate both approaches using a segmentation in the point cloud in both classes.

The exercise is nice, but the contribution is not very relevant. The lack of originality is my main concern with this work.

Some important issues:

1. The data set is not public, so there is no possibility to reproduce the results or to apply them to other data sets.

2. The results only show the benefits obtained by a comparison with the original DGCNN. I miss a more general study with other state of the art solutions.

3. The data set was manually labeled. I suggest to offer this openly to other researchers.

4. Authors mention the high computational cost of the method. There is no results about this question.

5. The features used as input: coordinates, roughness and elevation differences are standard ones, there is no novelties in them.

6. Point intensities are not used, but there is not a real reason for that decision. Authors mention the disparities in values, but this can be somehow solved.

7. Some minor mistakes that show authors did not took care to the final version of the paper at all:

- Section 2.4 in line 266 is not well labeled.

- Lines 169-170 are repeated.

- Figure 8 is not labeled. There are two figures 10.

Comments on the Quality of English Language

The quality of the language is good enough.